# Youth partnership in suicide prevention research: moving beyond the safety discourse

Maria Michail ,[1] Jamie Morgan,[1] Anna Lavis[2]

[1]Institute for Mental Health, School of Psychology, University of Birmingham, Birmingham, UK
[2]Institute of Applied Health Research, College of Medical and Dental Sciences, University of Birmingham, Birmingham, UK

**Correspondence to**
Dr Maria Michail;
M.Michail@bham.ac.uk

## ABSTRACT

**Objective** In this communication article, we discuss coproduction in suicide prevention research, with an emphasis on involving young people. We critically reflect on the lessons we have learned by working alongside young people, and how these lessons may be useful to other research teams.

**Summary** The meaningful involvement of young people in the design, implementation and translation of mental health research has received significant attention over the last decade. For most funding bodies, the involvement of patients and the public in the planning and delivery of research is advised and, in many cases, mandatory. When it comes to suicide prevention research, however, things are slightly different in practice. Involvement of young people in suicide prevention research has often been considered a controversial, unfeasible and even risky endeavour. In our experiences of working in this field, such concerns are expressed by funders, Higher Education Health and Safety committees and practitioners. By presenting an example from our research where the involvement of young people as experts by experience was integral, we highlight key lessons learnt that could maximise the potential of youth partnership in suicide prevention research. These lessons take on particular importance in mental health research against the background of long-entrenched power differences and the silencing of service user voices. Professional knowledge, obtained through education and vocational training, has historically taken priority over experiential knowledge obtained through lived experience, in psychiatric practice and research. Although this hierarchy has widely been challenged, any account of coproduction in mental health research is positioned against that background, and the remnants of those inequitable power relationships arguably take on greater resonance in suicide prevention research and require careful consideration to ensure meaningful involvement.

**Conclusion** We conclude that progress in suicide research cannot be fulfilled without the meaningful involvement of, and partnership with, young people with lived experience.

## INTRODUCTION

Involvement, coproduction, codesign—does terminology matter?

*Involvement* has been described as 'an evolving concept in the mental healthcare literature'[1] and is a term that encompasses service user involvement in both mental healthcare and research. In terms of the former, moves towards involving service users in their own care and decision-making ensued from both deinstitutionalisation and the political and philosophical critiques of psychiatry that emerged through the 1960s and 1970s in Europe and North America. These moves challenged both widespread institutionalisation and the biomedical model of mental illness. Service user involvement is now widely valued in health and social care.[2] It is key across a range of areas, from an individual's own care and decision-making, to service design, delivery and evaluation, policy-making, and research across a range of settings and topic areas. *No research about us without us* is a phrase commonly used to highlight the need for youth voices to be at the front and centre of research that has the potential to affect them, their families and the services or treatment they might receive, and is formalised in INVOLVE's[3] definition of involvement as 'research being carried out 'with' or 'by' non-professionals rather than 'to', 'about' or 'for' them.'

Against this background, the terms coproduction and codesign in applied health research are often used interchangeably, but it is key to note that both fall at one end of what has been termed a continuum of involvement.[4] At one end of this continuum is 'tokenism when a few service users might be asked to comment on or react to an agenda, project or document which has already been developed' (ibid.). The other end of the continuum is 'characterised by a different approach and ideology in which clients' expertise and knowledge is valued and an alternative discourse is recognised where service users may be involved in their own organisations in which they support each other and promote their shared agenda' (ibid.). This latter is where coproduction and

codesign fall, and is underpinned by an emphasis on the value of experiential knowledge, recognising this as being on an equal footing with professional knowledge. Given the history of silencing and disempowerment of mental health service users that is now widely acknowledged, this 'can be seen as an explicit call for persons with mental health diagnoses to exercise their rights as citizens to participate in the development, execution, and outcomes of research that directly affects them'[5]; and it is this that has led service user involvement to be described as an 'ethical imperative'.[6]

The term *coproduction* was initially conceptualised in the 1970s by Nobel Prize Winner Professor Elinor Ostrom,[7] a political scientist and economist. Ostrom used the term coproduction, within the context of public services, to highlight the lack of recognition of the contribution of consumers in public service delivery, and the impact this could have on the efficient running of public services. The term coproduction has since been used in many sectors including economics[8]; health policy and practice[9] and social care.[10] Within the context of mental health research, coproduction goes beyond mere 'collaboration'[11] and it is not simply something that is 'done,' to be implemented or added on; rather it should run throughout the full course of the research, actively shaping how this is done. Coproduction is a collaborative model of research, which emphasises the equality between, and combining of, different skills and experiences.[5] In turn, *codesign* is linked more closely to the concepts of user engagement and participatory design for the purposes of developing a new product, initiative, or service.[12]

Many attempts have been made to demarcate the terms coproduction and codesign, as well as involvement more broadly, but one could ask 'Does it really matter?'. A systematic review[13] of their varied definitions concluded that these terms are conceptualised in many different ways; and instead of focusing our efforts on finding a common language we should instead focus on defining the key principles and values underlying these terms; and how these could be meaningfully translated into every practice. In this communication article, we will use the terms involvement and coproduction.

## YOUTH INVOLVEMENT IN MENTAL HEALTH RESEARCH

Within the context of mental health research, the meaningful involvement of young people with lived or living experience in planning, delivering and implementing research has received widespread attention. Moves towards youth-led research, whereby young people as 'experts by experience' take on the role of agenda setters by defining priorities or directions of research, although slower, have been significant.[14] A recent example of youth-led research[15] was undertaken by the UK Research and Innovation-funded Transdisciplinary Research for the Improvement of Youth Mental Public Health mental health research network. The authors describe a UK-wide priority-setting exercise to identify public health

intervention-related research to support youth mental health, offering examples of good practice in coproduction, whereby young people take on the role of agents of change. The authors also reflect on the challenges of involving multiple stakeholders with different, and often competing, priorities and ensuring the voices of young people are not lost during the process of finding a shared language.[15]

The significant benefits resulting from such partnerships with young people to collaboratively develop and deliver high-quality mental health research[16–18] have been widely documented. For the research itself, benefits include improved data collection methods and recruitment strategies,[19] more relevant research objectives enhancing study acceptability,[19–21] and facilitating the translation of research findings into practice, therefore, creating real-world impact.[16] For young people with lived or living experience, recent research highlights several benefits, including drawing on their own experience to help and support others, gaining a sense of achievement through the impact of their involvement, as well as personal growth and capacity building.[18 22] It is key not to ignore the potential pitfalls of coproduction, such as various risks to both researchers and experts by experience[23] but also to recognise how current limitations may arise from existing contextual inequalities in academia and more broadly.[11] Against the background of the historically embedded power differentials between mental health service users and professionals, noted above, the possibility of reinforcing such unequal power relations across the research's various stakeholders needs recognising.[24 25] But so too does the danger of repeating these power imbalances by shutting young people out of involvement as experts by experience.

It is also clear that for young people with lived or living experience, coproduction can offer a way to make a difference, and even to reflect on one's own experiences.[26–28] Then, why is it that when it comes to suicide prevention research, the involvement of young people is often considered risky and unfeasible?

## YOUTH INVOLVEMENT IN SUICIDE PREVENTION RESEARCH

There is, first, a pervasive concern about the possibility of causing distress, and even suicidal thoughts and feelings.[29 30] Such concern is commonly raised by health research ethics committees and even researchers themselves[29 30] and relates to a deficit-based approach whereby young people with lived or living experience of self-harm or suicidal behaviour are perceived as 'vulnerable', needing 'protection from harm'. Associated with the concern of causing harm or distress are also concerns about researchers' own competency and capacity to support potentially distressed young people, and/or their families.[29] This mirrors a similar concern around young people taking part as research participants too—and yet, taking part in research does not inflict harm and young people cite important benefits.[31] So, why do we hold

onto this concern in such a way that it acts to preclude young people from being involved in suicide prevention research as experts by experience?

The safety of the young people is of paramount importance and the risk of involvement potentially causing distress needs to be acknowledged. It may be that there are limitations to involvement posed by legitimate safeguarding, depending on the topic of the research. However, we argue that the greatest risk to young people is not being heard and included. We risk leaning too heavily on a safety discourse in a way that allows us to not collaboratively find ways to undertake coproduction ethically and safely. We also run the risk of being unethical, exclusionary and paternalistic. These specific risks that arise from, or perhaps rather are *assumed to* arise from, suicide prevention research,[29 30] show the need to be mindful of not reproducing entrenched power imbalances between experts by experience on the one hand and 'professionals' on the other.

To undertake meaningful involvement in suicide prevention research, there is a need across the team—both young people and researchers—to sit with a certain level of risk. It is key that this is acknowledged at the outset of the research and collaboratively discussed. Before this stage, however, it is also necessary to rethink the way we approach risk in drawing the team together. There is a tendency in even the most ethical and equitable involvement to nevertheless exclude people who may be, or have recently been, suicidal. Given the subject matter and the need to ensure that the voices of young people who are or have been suicidal are placed central to our research praxis, we question whether this is ethical.

The need, therefore, to reflect critically on who is 'allowed' to become involved mirrors who is 'allowed' to take part in suicide prevention research as a participant too. In both contexts, if 'we' as the researchers set the bar in a way that is too risk averse, we will perpetuate the historical power differentials in mental health research and services noted above, and ignore, exclude, and disempower people whose voices should be listened to.

We provide below an example of coproduction in suicide prevention research drawn from our own research programme; highlighting opportunities for maximising the potential of partnerships between researchers and young people with lived or living experience. The case study has been informed by the second author (JM), a member of the Youth Advisory Group (YAG), at the Institute for Mental Health, University of Birmingham. The YAG consists of 18 young people aged between 18 and 25 with experience of or a strong interest in youth mental health, who work collaboratively with researchers to create and shape research into youth mental health including self-harm and suicide. The YAG is a diverse group of young people in terms of cultural representation, ethnic background, gender identity and sexual orientation.

## CASE STUDY

#MyGPguide—visiting your general practitioner: a guide for young people with lived experience of self-harm and suicidality

*#MyGPguide*[32] is an evidence-based resource codesigned with young people for young people with lived experience of self-harm and/or suicidal behaviour to prepare them for their consultation with their general practitioner (GP) offering practical support and guidance on: (1) what to consider before they visit their doctor, including preparing questions and booking an appointment; (2) how to manage the consultation, what their rights are with respect to confidentiality, what questions their doctor might ask them; and how to manage discussions about medication, safety planning and referral to mental health services; (3) What to do after the consultation and how their doctor can support them including signposting and accessing professional support.

Consultations about self-harm and suicidal thoughts and feelings can be challenging for both GPs and young people.[33 34] *#MyGPguide*, offers evidence-based, accessible and practical tips to facilitate the best consultation and support for young people at-risk of suicide.

This account of the creation of #MyGPguide will highlight and reflect on various issues that have been raised when considering the feasibility, value, ethics and safeguarding of coproduction in suicide prevention research.

Over a period of 6 months, 6 members from the Institute of Mental Health, University of Birmingham YAG worked in partnership with the project lead (first author) and the Institute's Youth Participation Lead (YPL) to codesign *#MyGPguide*. The role of the YPL is to facilitate meetings between researchers and the YAG; to process requests from researchers to involve the YAG in their work, and likewise help make it easy and comfortable for the YAG to be involved. With regards to safety, the YPL's role is important as they provide a constant for the YAG in their meetings with different researchers. By being a friendly face, and someone who is specifically there to facilitate the work of the YAG, the feelings that might accompany a young person in this sort of environment are effectively mitigated.

Consultations took place online due to COVID-19 restrictions at the time. The first consultation started with the YPL reminding everyone of the YAG terms of reference; and the project lead sharing key ground rules that would ensure that young people were involved in a safe, inclusive and respectful way throughout the process. The ground rules focused on (1) highlighting the importance of confidentiality (and its limits) and anonymity; (2) acknowledging and respecting different and complementary types of expertise; (3) listening respectfully to one another; (4) allowing others the time and space to share their story; (5) taking turns in speaking and being mindful of dominating discussions.

The first stage of this project was an 'initial ideas' meeting which involved brainstorming and discussing initial thoughts about the content and overall direction

that YAG members wanted the guide to take. This was a productive meeting and YAG members were given multiple chances to share their views. It is important to note that from the beginning and throughout the whole project YAG members were never asked to share or disclose personal experiences or events that might have been relevant to the project. Instead, they were encouraged to draw on their own experience to meaningfully inform the content of the guide. Moreover, YAG members were encouraged to contribute in different ways depending on their preference. This included sharing their views during the online consultations; writing down thoughts and emailing them to the project lead; or having a one-to-one chat with the project lead. For some YAG members, this flexible approach was invaluable as they found it difficult to share in the group some of the points that they wanted to make. These points related to their own and their friends' experience of visiting the GP and how they wanted these points conveyed in *#MyGPguide*. In response, the project lead expressed the importance of these points and their added value to *#MyGPguide*. The project lead subsequently described how these points would be incorporated in the guide ensuring that this process was carried out in a way that would accurately reflect YAG members' contribution that is, staying true to their voice.

Following the initial brainstorming session, the first draft of *#MyGPguide* was circulated to all YAG members involved in the project and they were encouraged to read the guide and send any feedback. Feedback could be given via different ways including highlighting or using track changes on the document, or simply drafting the feedback and emailing this back to the project lead. It was clear where YAG members' comments had been included in the draft guide alongside the evidence base.

To bring the guide to life, the project lead offered YAG members the opportunity to be involved in a few other ways. These included producing a quote relating to a specific aspect of the guide (eg, the importance of confidentiality) to be embedded within the guide, involvement in a video that would accompany the guide, and an opportunity to be involved in a webinar to promote *#MyGPguide*. Each of these opportunities were once again accompanied by the flexibility and inclusivity which characterised the whole process of involvement.

Subsequent iterations of *#MyGPguide* were shared with the YAG, and members were always encouraged to make additional contributions and provide feedback. It was clearly demonstrated how their contributions had informed the content and design of *#MyGPguide* as well as a dissemination plan targeting diverse audiences including young people, families, primary care practitioners and schools. YAG members were kept informed throughout the project about timelines, milestones including the release of *#MyGPguide*; and supported to take part in dissemination and outreach activities to promote the guide.

Throughout the process, YAG members were never asked to reveal personal information, and any that was given was given voluntarily, without prompt and in line with the YAG terms of reference. The discussions, either as part of the group or one-to-one, were equitable and balanced. The way young people's contributions influenced *#MyGPguide* was clearly demonstrated; and where feedback had not been taken on board the rationale was provided. For example, although important, some points and ideas might not have fit the form the guide was aiming to take, could not offer additional value, or did not speak to the target audience.

At no point were YAG members made to feel like their contribution would make or break the guide, and they were reassured that they could withdraw at any time. Crucially, though, YAG members were consistently reminded of how their involvement and any contribution they did make was important and valued.

Furthermore, the flexibility described in this account was one of the most central themes that ensured safety in the involvement of YAG. Although extremely important, safety does not simply refer to the idea of keeping someone safe from physical or psychological harm. The term safety within the context of suicide prevention research should also be taken to mean how comfortable and able a young person is to fully draw on their experience to inform the research they are involved in. The flexibility of involvement here was crucial to maintaining a comfortable environment. Indeed, if young people do not feel safe or comfortable, they will not be able to fully apply their experiences and expertise or they will avoid participating altogether. It is therefore in both the young persons and the researchers' interests to ensure safety during involvement.

## KEY LESSONS FOR YOUTH INVOLVEMENT IN SUICIDE PREVENTION RESEARCH

Young people were at the forefront of designing and disseminating *#MyGPguide*. Using the case study above and our previous research experience and experiential knowledge, we draw together key lessons underlying the optimal involvement of young people in suicide-related research.

First, having clearly defined and mutually agreed terms of reference and ground rules helps to generate and sustain inclusivity throughout the process of involving young people as experts by experience in suicide prevention research. As Clarke et al[35] highlighted, inclusivity is not simply about bringing together a group of stakeholders to share and exchange knowledge and views. Inclusivity is about generating and sustaining mutually agreed goals; sharing the same vision, or, as Clarke et al[35] described it 'a shared sense of identity and solidarity'.

Second, consulting with young people about potential ethical considerations, safety issues or stressors before and during coproduction and putting in place safety protocols tailored to young people's needs can ensure their

safe involvement. Such protocols could take the form of a wellness or safety plan where researchers and young people work together to identify potential stressors, strategies that the young person can use when they are feeling upset, a support person the researchers could contact if the young person wishes so. Having a one-to-one debriefing session with young people also offers them the opportunity to highlight things that worked well for them during a coproduction session and things that could be improved in future sessions. Examples of good practice using wellness plans when involving young people in suicide prevention research can be used as a guide.[36 37] A risk mitigation strategy when conducting participatory modelling workshops with young people with lived or living experience of self-harm or suicidal behaviour has also been published and can be used or adapted by other researchers in the field.[37] The first evidence-based guidelines for the safe and effective involvement of young people with lived and living experience in suicide research have recently been published and provide an invaluable resource for researchers.[38] We cannot eliminate risk, but we can put in place robust processes that help researchers mitigate risk. It is crucial to recognise that young people can draw on their own experiences to inform research (and, through this, practice and policy) without necessarily having to disclose or share those experiences. It is important to set the parameters of sharing at the outset of any research, both creating a safe space to do so and also emphasising that this is not an expectation. Having a dedicated YPL to support both young people and researchers during this process in a safe, productive and ethically sound way is key.

Third, access to appropriate training, supervision and reflective practice for researchers working in 'emotionally demanding research', including undertaking coproduction with young people with lived experience, is important in developing the knowledge and skills to mitigate potential risk issues but also in maintaining their own well-being.[39]

Finally, setting a culture of open, transparent and reciprocal communication between young people and researchers is key to the establishment of an equal relationship among all those involved in the coproduction and the continual challenging of hierarchies. The principle of transparent communication has consistently been highlighted by different participatory research approaches as key in fostering authenticity, collaboration and empowerment.[40 41] Coproduction should be a professional relationship between young people and researchers and, as in any other type of professional relationship, there should be clearly defined roles, clear objectives and an expectation of being listened to and valued.

It is our ethical and moral imperative, as researchers, to create a safe and inclusive environment that would empower the involvement of young people with lived or living experience of self-harm and suicidal behaviour in research that affects them, their care and quality of life. Progress in the field of suicide prevention cannot be achieved without the meaningful involvement of young people.

**Contributors** All authors conceptualised the paper. MM wrote the first draft of the manuscript. All authors were involved in further revisions and writing of the final version.

**Funding** While working on this Communication article, MM was funded by the European Union H2020-EU.1.3.2. MSCA-IF-GF—Global Fellowships (grant agreement ID: 101026065). JM was funded by the Institute for Mental Health, University of Birmingham.

**Competing interests** None declared.

**Ethics approval** Not applicable.

**Provenance and peer review** Not commissioned; externally peer reviewed.

**ORCID iD**
Maria Michail http://orcid.org/0000-0001-7380-3494

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
