## [Reviewer comments · BMJ Open]

ARTICLE DETAILS

TITLE (PROVISIONAL)	Youth partnership in suicide prevention research: moving beyond the safety discourse
AUTHORS	Michail, Maria; Morgan, Jamie; Lavis, Anna

VERSION 1 – REVIEW

REVIEWER	Kohlbeck, Sara Medical College of Wisconsin, Comprehensive Injury Center
REVIEW RETURNED	14-Aug-2023

GENERAL COMMENTS	I really enjoyed the opportunity to review this paper and appreciate the space that is being made for this dialogue. This is very important work for the field of suicide research. I have just a couple of minor suggested edits: 1. Rather than using the word "trigger" or "triggering", I would suggest alternate language such as "stressor" or "distressing". In the field of suicide research, the word triggering could be seen as insensitive.2. What is the author's viewpoint on following young people's lead on research topics to pursue? Does that have a place in this work as well? I would like to see some comment about this in this manuscript. Thank you again for the opportunity to review!
--

REVIEWER	Chavez, Noé Charles R Drew University of Medicine and Science, Social Sciences and Humanities
REVIEW RETURNED	15-Aug-2023

GENERAL COMMENTS	Thank you for the opportunity to offer my humble thoughts and suggestions. You provide excellent and insightful reflections on the importance of youth inclusion in mental health services. This is a very important communication article that encourages us to improve our partnership with young people and offers concrete insights and recommendations based on youth partnership projects on suicide prevention. We are living in times of great mental health challenges especially for youth and their authentic inclusion in all facets of research on mental health and specifically on suicide prevention is morally imperative, with clear benefits beyond solely focusing on the risks, for improving our mental health systems and support to youth. The following are some thoughts and suggestions for your paper: 1. Have you reviewed the extensive research literature in youth
---

	participatory action research (YPAR) or community-based participatory research (CBPR) with youth? These youth centered approaches have been applied in many educational, social, and health areas, including mental health research. There are of course various other approaches or terms used that encompass similar core principles of youth partnership and increasing equity and empowerment. You may want to review some of the YPAR or CBPR literature or other similar research that applies principles of youth-adult equitable partnerships in youth mental health for additional insights of how other concepts or approaches may intersect with yours and your conceptual thinking. You may consider citing a few articles from this literature that focus on YPAR or CBPR and mental health for youth to acknowledge the growing research done in this area and briefly in a few sentences summarize how it relates to your work. However, I applaud your work since there is very limited research specifically in YPAR or CBPR and suicide prevention. The general research in YPAR or CBPR and mental health may offer some new concepts that can aid in your efforts. Some potential articles you may consider include the following (some of these articles also include conceptualization of their “Co-Design” approach or generally principles in youth-adult partnerships, which may or may not overlap or relate to your approach but may offer some ideas). a. Dold, C. J., & Chapman, R. A. (2012). Hearing a voice: Results of a participatory action research study. Journal of Child and Family Studies, 21, 512-519. b. Ungar, M., McGrath, P., Black, D., Sketris, I., Whitman, S., & Liebenberg, L. (2015). Contribution of participatory action research to knowledge mobilization in mental health services for children and families. Qualitative Social Work, 14(5), 599-615. c. Ospina-Pinillos, L., Davenport, T., Mendoza Diaz, A., Navarro-Mancilla, A., Scott, E. M., & Hickie, I. B. (2019). Using participatory design methodologies to co-design and culturally adapt the Spanish version of the mental health eClinic: qualitative study. Journal of medical Internet research, 21(8), e14127. d. Larkin, M., Boden, Z. V., & Newton, E. (2015). On the brink of genuinely collaborative care: experience-based co-design in mental health. Qualitative health research, 25(11), 1463-1476. e. Vélez-Grau, C. (2019). Using photovoice to examine adolescents' experiences receiving mental health services in the United States. Health Promotion International, 34(5), 912-920. f. Ramey, H. L., & Rose-Krasnor, L. (2015). The new mentality: Youth–adult partnerships in community mental health promotion. Children and Youth Services Review, 50, 28-37. 2. Can you provide information about the demographics of the youth you partnered with, such as their race/ethnicity, immigrant status, gender, sexual orientation or other important social identities? Were you intentional in your youth inclusion process to be mindful of other intersecting social identities along with their ‘youth identity’ and ‘youth culture’? In what ways did you mindfully consider this aspect of inclusion which in itself has other power dynamics at play that may shape the youth-adult partnership or youth-peer relationships? 3. I am not sure about the space, formatting or standard requirements for a ‘communication article type’ – would it be possible if space is available to provide a simple visual diagram of some type to capture more easily your main stages of the process of youth engagement and partnership to develop your #MyGPGuide?
--	---

REVIEWER	Marraccini, Marisa E.
-----------------	-----------------------

	The University of North Carolina at Chapel Hill
REVIEW RETURNED	24-Aug-2023

GENERAL COMMENTS	I read this commentary with interest and believe it provides innovative ideas with the potential to inform co-design and co-production with an overlooked population: youth with suicide-related thoughts and behaviors. I particularly appreciated that the reflections were from a youth leader in collaboration with researchers. I found the commentary well written. I offer the following suggestions in order to strengthen the paper and support transferability of the commentary:  - Consider adding more specific descriptions of the benefits of co-design/co-production to youth in the introduction; -The commentary assumes concerns for co-design with youth with suicide-related risk, but includes no citations describing criticisms/concerns for this methodology with this population. Describing specific concerns and including citations, then linking this to ways of handling/overcoming concerns, would greatly improve practical implications of this commentary. -A more detailed explanation of recruitment and specific approaches used for co-design approaches would also help other researchers better learn how to apply such an approach; -More specific descriptions of how concerns for contagion effects were prevented and/or handled and how stressors/triggers emerging during co-design were handled (the authors describe planning for them at the onset, but did not include a description or case example of how they were handled) would also increase practical implications.
--

VERSION 1 – AUTHOR RESPONSE

Reviewer: 1

Dr. Sara Kohlbeck, Medical College of Wisconsin

Comments to the Author:

I really enjoyed the opportunity to review this paper and appreciate the space that is being made for this dialogue. This is very important work for the field of suicide research. I have just a couple of minor suggested edits:

1. Rather than using the word "trigger" or "triggering", I would suggest alternate language such as "stressor" or "distressing". In the field of suicide research, the word triggering could be seen as insensitive.

Response: We would like to thank the reviewer for this suggestion. We have now replaced the word “trigger” and “triggering” with more acceptable language (lines 150, 163, 308, 311).

2. What is the author's viewpoint on following young people's lead on research topics to pursue? Does that have a place in this work as well? I would like to see some comment about this in this manuscript.

Response: We have now included a short section (lines 114-124) where we comment on laudable developments in youth-led research. Although moves towards including young people as research agenda-setters are slow, there have been some significant developments

in this area. We present an example of good practice where the authors reflect on challenges of involving multiple stakeholders with different, and often competing, priorities and ensuring the voices of young people are not lost during the process of finding a shared language.

Reviewer: 2

Comments to the Author:

Thank you for the opportunity to offer my humble thoughts and suggestions. You provide excellent and insightful reflections on the importance of youth inclusion in mental health services. This is a very important communication article that encourages us to improve our partnership with young people and offers concrete insights and recommendations based on youth partnership projects on suicide prevention. We are living in times of great mental health challenges especially for youth and their authentic inclusion in all facets of research on mental health and specifically on suicide prevention is morally imperative, with clear benefits beyond solely focusing on the risks, for improving our mental health systems and support to youth. The following are some thoughts and suggestions for your paper:

1. Have you reviewed the extensive research literature in youth participatory action research (YPAR) or community-based participatory research (CBPR) with youth? These youth centered approaches have been applied in many educational, social, and health areas, including mental health research. There are of course various other approaches or terms used that encompass similar core principles of youth partnership and increasing equity and empowerment. You may want to review some of the YPAR or CBPR literature or other similar research that applies principles of youth-adult equitable partnerships in youth mental health for additional insights of how other concepts or approaches may intersect with yours and your conceptual thinking. You may consider citing a few articles from this literature that focus on YPAR or CBPR and mental health for youth to acknowledge the growing research done in this area and briefly in a few sentences summarize how it relates to your work. However, I applaud your work since there is very limited research specifically in YPAR or CBPR and suicide prevention. The general research in YPAR or CBPR and mental health may offer some new concepts that can aid in your efforts. Some potential articles you may consider include the following (some of these articles also include conceptualization of their “Co-Design” approach or generally principles in youth-adult partnerships, which may or may not overlap or relate to your approach but may offer some ideas).

a. Dold, C. J., & Chapman, R. A. (2012). Hearing a voice: Results of a participatory action research study. *Journal of Child and Family Studies*, 21, 512-519.

b. Ungar, M., McGrath, P., Black, D., Sketris, I., Whitman, S., & Liebenberg, L. (2015). Contribution of participatory action research to knowledge mobilization in mental health services for children and families. *Qualitative Social Work*, 14(5), 599-615.

c. Ospina-Pinillos, L., Davenport, T., Mendoza Diaz, A., Navarro-Mancilla, A., Scott, E. M., & Hickie, I. B. (2019). Using participatory design methodologies to co-design and culturally adapt the Spanish version of the mental health eClinic: qualitative study. *Journal of medical Internet research*, 21(8), e14127.

d. Larkin, M., Boden, Z. V., & Newton, E. (2015). On the brink of genuinely collaborative care: experience-based co-design in mental health. *Qualitative health research*, 25(11), 1463-1476.

e. Vélez-Grau, C. (2019). Using photovoice to examine adolescents' experiences receiving mental health services in the United States. *Health Promotion International*, 34(5), 912-920.

f. Ramey, H. L., & Rose-Krasnor, L. (2015). The new mentality: Youth–adult partnerships in community mental health promotion. *Children and Youth Services Review*, 50, 28-37.

Response: We would like to thank the reviewer for this suggestion. We agree that there is a plethora of approaches to youth partnership in mental health research and although these approaches might be implemented differently, core principles and values underlying these approaches are largely similar:

- **Empowerment of young people: We added a section on youth-led research (i.e., young people as agenda-setters) as a way of empowering young people and illustrate the importance of empowerment through a priority-setting example using PAR (lines 114-124).**
 - **Ethical considerations and risk: We have drawn upon a previous study (lines 315-317) using participatory co-design process in suicide prevention research to shape and inform our reflections on managing risk and safety when partnership with young people. We also refer to our own risk mitigation strategy (Michail et al, 2023) using participatory modelling workshops with young people with lived experience of self-harm and/or suicidal behaviour (lines 317-320).**
 - **We have reflected and expanded on the benefits of youth partnership reported by different youth-centred approaches including YPAR (lines 127-133) supported by relevant citations.**
 - **Transparent communication: We reflect on similarities between our approach and other PAR approaches (Kennedy et al, 2009; Doucet et al, 2022) on the importance of open and transparent communication in fostering authenticity, collaboration and empowerment (lines 337-339).**
2. Can you provide information about the demographics of the youth you partnered with, such as their race/ethnicity, immigrant status, gender, sexual orientation or other important social identities? Were you intentional in your youth inclusion process to be mindful of other intersecting social identities along with their 'youth identity' and 'youth culture'? In what ways did you mindfully consider this aspect of inclusion which in itself has other power dynamics at play that may shape the youth-adult partnership or youth-peer relationships?

Response: We would like to thank the reviewer for raising this comment. Although we do not have YAG members' consent to publish their demographic information, we have highlighted in the manuscript (lines 192-194) that the YAG is a diverse group of young people in terms of cultural representation, ethnic background, gender identity and sexual orientation. We made a concerted effort to maximise opportunities for wider engagement with young people by seeking collaborative partnerships with Birmingham-based youth engagement organisations and establishing a larger online presence within the youth community through social media platforms. Power and social dynamics within such a diverse group are inevitable and throughout the case study we provided examples of how group dynamics can be effectively managed: 1) having clearly defined ground rules (lines 224-231), 2) ensuring all voices are heard by taking YAG members' feedback into consideration when designing *#MyGPguide* (lines 253-256, 265-268), 4) ensuring circular flow of knowledge i.e. checking and clarifying YAG members' understanding (lines 276-280) and 5) empowering participation by validating YAG members' contributions (lines 283-284).

3. I am not sure about the space, formatting or standard requirements for a 'communication article type' – would it be possible if space is available to provide a simple visual diagram of some type to capture more easily your main stages of the process of youth engagement and partnership to develop your *#MyGPguide*?

Response: We would like to thank the reviewer for this suggestion. However, after careful consideration we decided to not provide a visual diagram to capture the process of developing the guide. As the case study is reflective, and not necessarily descriptive of the different stages of co-production, we are unsure about the value such a diagram would add. The case study is used as an example of reflecting on various issues in relation to the feasibility, value, ethics and safeguarding of co-production in suicide prevention research. Adding a diagram on

the main stages of developing *#MyGPguide*, although practical, it would dilute the focus of the key messages communicated in the reflective piece.

Reviewer: 3

Dr. Marisa E. Marraccini, The University of North Carolina at Chapel Hill

Comments to the Author:

I read this commentary with interest and believe it provides innovative ideas with the potential to inform co-design and co-production with an overlooked population: youth with suicide-related thoughts and behaviors. I particularly appreciated that the reflections were from a youth leader in collaboration with researchers. I found the commentary well written.

I offer the following suggestions in order to strengthen the paper and support transferability of the commentary:

- Consider adding more specific descriptions of the benefits of co-design/co-production to youth in the introduction;

Response: We would like to thank the reviewer for this suggestion. We have now added more specific descriptions of the benefits of involving young people in mental health research with supporting citations (lines 127-133). These benefits refer to young people themselves and the research.

-The commentary assumes concerns for co-design with youth with suicide-related risk but includes no citations describing criticisms/concerns for this methodology with this population. Describing specific concerns and including citations, then linking this to ways of handling/overcoming concerns, would greatly improve practical implications of this commentary.

Response: We have now specified two of the most commonly raised concerns in relation to youth involvement in co-design for suicide prevention research supported by relevant citations (lines 150-157) and have provided recommendations on how to address these concerns (lines 315-322; 330-333).

-A more detailed explanation of recruitment and specific approaches used for co-design approaches would also help other researchers better learn how to apply such an approach;

Response: We would like to thank the reviewer for this comment. Recruitment approaches and strategies used for co-design to ensure diversity and wide representation is an important topic. After careful consideration, we respectfully disagree with the reviewer's suggestion to add a detailed explanation and specific approaches used for co-design for two reasons:

1. This topic has already been widely covered and documented in the field of health, mental health and beyond (e.g., Chauhan et al, 2021). The UK National Institute for Health Research, for example, provides specific recommendations and strategies about working collaboratively with individuals from marginalised communities to co-design research: How to generate inclusive and diverse mental health patient and public involvement and engagement. We do not have any further insights or reflections to share that would add value to the established literature on this topic; on the contrary, we would be running the risk of being repetitive.
2. We believe it is outside the remit of the specific commentary to provide a detailed explanation or a step-by-step guide on recruitment approaches for co-production and doing so would divert from the focus of the commentary which is centered around the need to move beyond the safety/risk discourse, which is still very prevalent, when it comes to youth partnership in suicide prevention research. The commentary aims to challenge some of the myths around this topic and to this effect we have provided specific recommendations and reflections from our own research experience and experiential knowledge.

-More specific descriptions of how concerns for contagion effects were prevented and/or handled and how stressors/triggers emerging during co-design were handled (the authors describe planning for them at the onset, but did not include a description or case example of how they were handled) would also increase practical implications.

Response: We would like to thank the reviewer for this comment and the opportunity to clarify that no concerns, risk, or safety issues were identified or reported during the co-design of #MyGPguide. Therefore, although there were procedures in place to address potential concerns or safety issues, we did not have to use these. We do provide guidance, however, on how to address potential concerns arising in future research (lines 307-322). We have now added examples of good practice where wellness plans and risk mitigation strategies have been previously used in suicide prevention research and encourage researchers to adopt or adapt these in line with the needs and demands of their own research.

VERSION 2 – REVIEW

REVIEWER	Kohlbeck, Sara Medical College of Wisconsin, Comprehensive Injury Center
REVIEW RETURNED	22-Sep-2023

GENERAL COMMENTS	Thank you for your revisions. The paper is much improved, and, in my opinion, it is suitable for publication. Thank you for your contribution in this space. I learned from reading your paper.
---

REVIEWER	Chavez, Noé Charles R Drew University of Medicine and Science, Social Sciences and Humanities
REVIEW RETURNED	05-Oct-2023

GENERAL COMMENTS	The authors provided a clear and thoughtful response to reviewers and integrated recommendations where they saw appropriate. Two of the reviewers recommended any a visual diagram or a section offering more detailed information regarding the process of the stages of their research, such as youth recruitment and engagement. The authors provided a thoughtful justification for why they did not agree they should include more of this information. I think their rationale is justified given that their paper is specifically a commentary piece and their focus are on their reflections of the substantive issues tied to youth partnership and inclusion in suicide prevention research. I think in this case the editors, based in part on the formatting/structure of commentaries, should determine if the authors' rationale is indeed valid. Overall, the updates to the paper strengthened it. The paper would be an important contribution to the relevant literature. It particularly adds knowledge about the importance of inclusion of youth voices in the essential and difficult area of suicide prevention research. Much needed writing to convey that the benefits outweigh the risks of youth inclusion.
---

REVIEWER	Marraccini, Marisa E. The University of North Carolina at Chapel Hill
REVIEW RETURNED	05-Oct-2023

GENERAL COMMENTS	Thank you for the opportunity to review this important manuscript a second time. I found the paper to be strengthened and appreciate the authors justification for responding to or not responding to reviewer comments. I believe this is an impactful, well-written, and much needed paper and therefore strongly recommend accepting it
--

	for publication. Congratulations on contributing such an important work to the field.
--	---